# Transformer-Based User Alignment Model across Social Networks

**Tianliang Lei** [1], **Lixin Ji** [1,2], **Gengrun Wang** [1], **Shuxin Liu** [1,*] , **Lan Wu** [1] **and Fei Pan** [1]

1. Institute of Information Technology, Information Engineering University, Zhengzhou 450002, China; leitianliang1@163.com (T.L.)
2. Songshan Laboratory, Zhengzhou 450018, China
* Correspondence: liushuxin11@126.com

**Abstract:** Cross-social network user identification refers to finding users with the same identity in multiple social networks, which is widely used in the cross-network recommendation, link prediction, personality recommendation, and data mining. At present, the traditional method is to obtain network structure information from neighboring nodes through graph convolution, and embed social networks into the low-dimensional vector space. However, as the network depth increases, the effect of the model will decrease. Therefore, in order to better obtain the network embedding representation, a Transformer-based user alignment model (TUAM) across social networks is proposed. This model converts the node information and network structure information from the graph data form into sequence data through a specific encoding method. Then, it inputs the data to the proposed model to learn the low-dimensional vector representation of the user. Finally, it maps the two social networks to the same feature space for alignment. Experiments on real datasets show that compared with GAT, TUAM improved ACC@10 indicators by 11.61% and 16.53% on Facebook–Twitter and Weibo–Douban datasets, respectively. This illustrates that the proposed model has a better performance compared to other user alignment models.

**Keywords:** user alignment; cross-social networks; data mining; machine learning

## 1. Introduction

With the development and popularization of the Internet and related mobile devices, social networks became an important part of people's lives. A large number of people share their lives, work, or exchange information with each other on multiple social networking platforms to meet their different social requirements. For example, a user can either share their daily life on Facebook or express their opinion on Twitter. Although multiple social networks greatly enrich people's lives, there are many problems involving the joint research and analysis of multiple networks, and the problem of user alignment across social networks came into being. This problem aims to combine multiple social networks to analyze the relationships between nodes and construct a high-quality cross-social network user alignment model to connect the same person on different social networks. This model can be widely used in various fields, such as cross-network recommendation [1], cross-domain information diffusion [2,3], link prediction [4], and network dynamics analysis [5].

For user alignment across social networks, some researchers proposed a user alignment method based on user location information. Riederer et al. [6] utilized the rich user position information in location-based social networks (LBSNs) to propose a POIS algorithm, which started from the user's trajectory data, analyzed the similarity of user pairs, and designed a general and self-tunable algorithm to align users between two LBSNs. Chen et al. [7] used kernel density estimation (KDE) to alleviate the data sparsity in measuring user similarity, and further organized location data based on the structure of the grid. Then, they pruned and reduced the search space to improve the efficiency of user alignment.

At the same time, some researchers proposed a large number of cross-social network user alignment methods based on user profile information [8–11]. This information includes

a variety of profile information about users in two networks, such as usernames, educational experiences, cities of residence, and personal descriptions. It showed that the cross-social network user alignment model based on user profile information is feasible and effective for some social networks. Zhang et al. [11] proposed MOBIUS, where user similarity between different social networks is measured by extracting some user characteristics, such as prefixes, suffixes, the rarity of usernames, and user habits. Zhao et al. [12] proposed a BP neural network mapping for social network alignment, which used the BP neural network to obtain the mapping between user name vectors across social networks, changed the classification problem into a mapping problem between vectors, and improved the accuracy of social network alignment. However, in the actual social network scenario, the user's profile information is difficult to obtain, which involves the user's privacy. Many user profile information cannot be accessed, and many users will imitate others or forge personal information for various purposes, making the user alignment method based on user profile information difficult to work.

Aiming at the user alignment model based on user profiles, some researchers believed that the same person has a similar structure in different social networks, so they proposed cross-social network user alignment methods based on the user's local social structure [13,14], and user-based local and global social structure [15]. However, the actual situation is that due to the different service functions of different social networks, the same person has different social network structures in different social networks. In this regard, some researchers applied network embedding representations to cross-social network user alignment. Feng et al. [16] proposed a hypergraph neural network (HGNN) framework for data representation learning, which encodes higher-order data correlations in the hypergraph structure, and a hyperedge convolution operation to process these correlations achieves good results. On this basis, Chen et al. [17] proposed a multi-layer graph convolutional network (MGCN) that jointly considers the local network structure and hypergraph structure. In addition, a two-stage spatial coordination mechanism is proposed to efficiently align users across different large-scale social networks. Although user alignment models based on network embedding were proven to be effective, the problem of "too close" representation of network embedding is also unavoidable, which greatly affects the accuracy of the model. Yan et al. [18] introduced pseudo-anchors to make the distribution of user embedding representations more uniform and proposed a meta-learning algorithm to guide the update of pseudo-anchors, which effectively solved the problem that the network embedding representations are too close. In sparse networks, user network structural similarities are small and difficult to identify. Li et al. [19] proposed a triple-layer attention mechanism-based network embedding (TANE) method, which learns latent structural information by using the weighted structural similarity of the first-order and second-order neighbors to reduce network sparsity, and fully mines the network structure to identify users. He et al. [20] proposed a heuristic algorithm based on the attention mechanism HDyNA, which obtained the local importance weight of new nodes in a single network through the attention mechanism. It used the anchor node as supervision information and heuristically learned the local influence driven by the alignment task of new nodes to improve the performance of model alignment across dynamic networks. To reduce the expression of noise edges for structural consistency across social networks, Liu et al. [21] proposed a network structure denoising framework, which learned the user network topology and removes noise edges by iterative learning through a parameter sharing encoder and graph neural network (GNN) to improve the structural similarity across networks. Zheng et al. [22] considered the influence of distribution differences between different networks on model performance, and a periodically consistent adversarial mapping model (CAMU) was proposed, which learned the mapping function across potential representation spaces and solved the representation distribution difference through adversarial training between the mapping function and discriminator. In addition, periodic consistency training can alleviate the overfitting problem and reduce the number of labeled users required. The comparisons of the user alignment models are listed in Table 1. Most of the

existing models did not assign weight and some of them assigned local weight and had the problem of "over-smooth".

**Table 1.** The comparisons of the existing user alignment models.

| Methods | References | User Location | User Profile | Network Structure | Weight Assignment | | Open Datasets | Over-Smooth |
|---|---|---|---|---|---|---|---|---|
| | | | | | Local | Global | | |
| Non-deep learning | [6] | ✓ | - | - | ✗ | ✗ | ✗ | - |
| | [7] | ✓ | - | - | ✗ | ✓ | ✓ | - |
| | [11] | - | ✓ | - | ✗ | ✗ | ✗ | - |
| Deep learning | [12] | - | ✓ | - | ✗ | ✓ | ✓ | ✗ |
| | [16] | - | - | ✓ | ✗ | ✓ | ✓ | ✓ |
| | [17] | - | - | ✓ | ✗ | ✓ | ✓ | ✓ |
| | [18] | - | - | ✓ | ✗ | ✓ | ✓ | ✓ |
| | [19] | - | - | ✓ | ✓ | ✓ | ✓ | ✓ |
| | [20] | - | - | ✓ | ✓ | ✓ | ✓ | ✓ |
| | [21] | - | - | ✓ | ✗ | ✓ | ✓ | ✓ |
| | [22] | - | - | ✓ | ✗ | ✓ | ✓ | ✓ |
| | **TUAM** | - | - | ✓ | ✗ | ✓ | ✓ | ✗ |

However, the above methods based on GNN to mine user network structure information will appear "over-smooth" with the deepening of the number of network layers. That is to say, the characteristics of all nodes in the same connected component tend to be consistent after multiple convolution operations, resulting in an extreme decrease in the effect of the model. Inspired by Yin et al. [23], graph structure information is encoded into the model via Transformer. To fill the research gaps, a Transformer-based user alignment model (TUAM) across social networks is proposed in this paper, which accurately imports the graph structure information into the model through three encoding methods, calculates the semantic similarity between cross-social network nodes, and obtains the accurate expression of network nodes to solve the problem of "over-smooth".

The main conclusions and novelties of this paper can be summarized as follows: First, a Transformer-based user alignment model (TUAM) is proposed to model node embeddings in social networks. This method transforms the graph structure data into a sequence data type that is convenient for Transformer learning through three novel graph structure encoding methods, which effectively avoids the phenomenon of "over-smooth" of GNN. Second, TUAM can assign the weight of different users' influence and network structure, accurately model the embedding vector of users, and improve the accuracy of social network alignment. Third, experiments on real datasets Facebook–Twitter and Weibo–Douban show that the results of the proposed model are superior to existing models.

## 2. Methodology

### 2.1. Development of Transformer

With the advent of better computer hardware, such as graphic processing units (GPUs), and word embedding methods, such as Word2Vec and Glove, deep learning models, such as convolutional neural networks (CNN) and recurrent neural networks (RNN) gained wider use in building natural language processing (NLP) systems. However, word-by-word processing of RNN limits computational efficiency, so Vaswani et al. [24] proposed a deep learning model Transformer based on self-attention, which contains layers of stacked encoders and decoders that allow it to learn complex linguistic information. In the field of NLP, the ability of Transformer and self-supervised learning is combined to develop a Transformer-based pre-training language model (T-PTLM). A generative pre-trained Transformer (GPT) is based on T-PTLM and developed at the Transformer decoder layer. BERT [25] is the first T-PTLM developed based on the Transformer encoder layer. The study by Kaplan et al. [26] showed that the performance of T-PTLM could be improved simply by increasing the size of the model, and the results drive the large-scale development of T-PTLM, such as GPT-3 [27], PANGU [28], and GShard [29]. Some of which can contain billions of parameters, and switch-Transformers [30] contain trillions of parameters.

With the success of T-PTLM, they are also used in other fields, such as finance, biomedicine, computer vision, etc. Dosovitskiy et al. [31] proposed the Transformer-based

ViT model, which is simple, effective, and extensible, and became a milestone work in the application of Transformer in the field of computer vision. However, the computational complexity of the self-attention module of the ViT model is very high. Therefore, Liu et al. [32] proposed the Swin Transformer model, which not only adopts a pyramidal hierarchical structure, but also proposes a linear complexity attention calculation, which is very powerful in downstream tasks.

### 2.2. Transformer Model

Transformer is a deep learning model based entirely on the self-attention mechanism, which replaces long short-term memory (LSTM) with the attention mechanism. Transformer abandons the inherent mode of the previous traditional encoder–decoder model that must be combined with CNN or RNN, which not only reduces the computation complexity and improves parallel efficiency, but also is higher in accuracy and performance than the popular RNN.

The Transformer consists of two parts: encoders and decoders. Each layer of the encoder has two sublayers, one is the multi-head attention mechanism, and the other is the position fully connected feed-forward network. Each sublayer uses a residual connection and layer normalization. Unlike encoders, decoders insert a third sublayer in addition to two sublayers in the encoder layer, which performs a multi-head attention mechanism learning model on the output of the encoder.

The inputs of the self-attention module are represented as $X = [x_1^T, \ldots, x_N^T] \in \mathbb{R}^{N \times d}$, where $d$ is the dimension of input features. The corresponding $Q$, $K$, and $V$ can be calculated using the input matrix $X$ and the linear array matrix $W_Q \in \mathbb{R}^{d \times d_q}$, $W_K \in \mathbb{R}^{d \times d_k}$, and $W_V \in \mathbb{R}^{d \times d_v}$, where, $d_q$, $d_k$, and $d_v$ are the corresponding feature dimensions, respectively. Assume that $d = d_q = d_k = d_v$, and they can be calculated as [24]

$$Q = XW_Q, K = XW_K, V = XW_V. \tag{1}$$

The output of the self-attention module is calculated as follows:

$$Attention(Q, K, V) = softmax\left(\frac{QK^T}{\sqrt{d_k}}\right)V. \tag{2}$$

### 2.3. Problem Definition

Cross-social network user alignment is also known as user identity linkage [33]. It is different from predicting the connection relationship between two or more different users on a single network [34–37]. Instead, it is to find correspondence between different identities of the same user in multiple social networks. In this section, we first introduce some necessary definitions and then give a formal definition of user alignment across social networks.

**Definition 1.** *Social networks: Represented as $G = (V, E, X)$, where $V = \{v_i | i = 1, \ldots, N\}$ is a set of user nodes, $E = \{e_{ij} = (v_i, v_j) | v_i \in V, v_j \in V\}$ is a set of user edges, $e_{ij} = (v_i, v_j)$ represents the connection status of node i and node j. If node i has a connection with node j, $e_{ij} = 1$, otherwise $e_{ij} = 0$. $X = \{x_i | i = 1, \ldots, N\}$ denotes a set of user feature vectors, $x_i$ is the feature vector of the ith user.*

**Definition 2.** *Source social networks and target social networks: the problems of cross-social network user alignment between two social networks are mainly studied, so the two social networks are named the source social networks $G_S$ and the target social networks $G_T$, where the source social network is $G_S = (V_S, E_S, X_S)$ and the target social network is $G_T = (V_T, E_T, X_T)$.*

**Definition 3.** *Anchor user: Given source social network as $G_S = (V_S, E_S)$ and target social network as $G_T = (V_T, E_T)$, where the user set belonging to the same person is the anchor user set*

$T = \left\{ \left( u^s, v^t \right) \middle| u^s \in V_S, v^t \in V_T \right\}$ *and* $e_{ij}{}' = \left( u_i^s, v_j^t \right)$ *represents the anchor link between* $G_S$ *and* $G_T$*. The cross-social network user alignment problem is essentially equivalent to the anchor link prediction problem between* $G_S$ *and* $G_T$*.*

## 3. A Transformer-Based User Alignment Model

The proposed TUAM includes three encoding design methods to aim at the "over-smooth" problem in the current cross-social network user alignment problem based on GNN. The overall framework of the proposed model is shown in Figure 1.

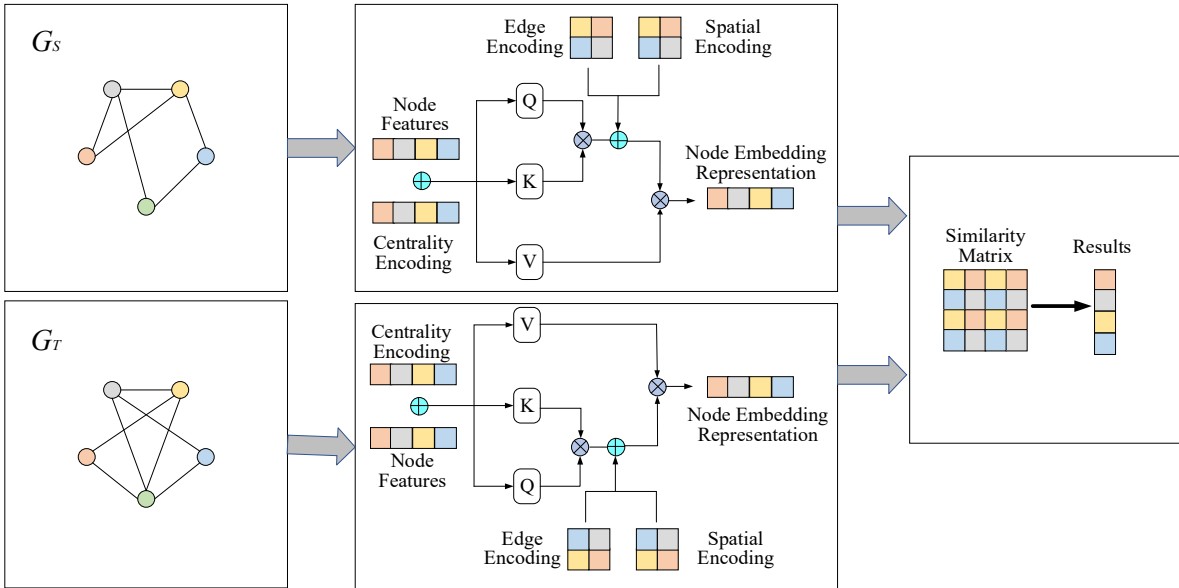

**Figure 1.** The overall framework of TUMA across social networks.

The proposed model represents the node embeddings of different social networks to the same vector space, and finally obtains the alignment results according to the node embedding vector similarity matrix.

### 3.1. Centrality Encoding

Node centrality is used to measure the importance of nodes in the graph and it is important information in the graph structure, such as celebrities who are followed by everyone is an important factor in predicting the trend of social networks [38,39]. However, this information is often ignored in previous graph convolution operations and is very valuable.

In the proposed model, centrality encoding takes degree centrality, which is one of the standard centrality measures, as an additional signal to the model, and assigns each node two real-valued embedding vectors based on its in-degree and out-degree. Since each node is centrally encoded, it can be added to the node feature as the input of the proposed model. The important information of nodes is input to the model through centrality encoding, and the semantic correlation and node importance between nodes are obtained through the attention mechanism. Centrality encoding formulas is [23]

$$h_i^{(0)} = x_i + z_{\deg^-(v_i)}^- + z_{\deg^+(v_i)}^+, \tag{3}$$

where $z^-, z^+ \in \mathbb{R}^d$ are the corresponding learnable embedding vectors with dimension $d$ in directed graphs of in-degree $\deg^-(v_i)$ and out-degree $\deg^+(v_i)$, respectively. In undirected graphs, in-degree $\deg^-(v_i)$ and out-degree $\deg^+(v_i)$ can be unified to $\deg(v_i)$.

### 3.2. Spatial Encoding

One advantage of the Transformer model over GNN is its global receptive field. In the Transformer layer, the attention mechanism can focus on and process information at any location. Notably, the position dependencies in individual node locations or the encoding layer need to be clear. Therefore, for sequence data, there are two ways to indicate node position information. One is to assign an absolute position, that is, absolute position encoding, and the other is to encode the relative distance of any two nodes in the Transformer layer, that is, relative position encoding. However, nodes are not sequential in graphs, they can be in multi-dimensional space and connected by edges. To obtain the structural information of the coding diagram in the model, this paper adopts a novel spatial encoding method. To measure the spatial relationship between two nodes $v_i$ and $v_j$ in the graph, a function $\phi(v_i, v_j) : V \times V \to \mathbb{R}$ is designed to represent the shortest path distance between $v_i$ and $v_j$. If there is no connection relationship between the two nodes, a special value is set for each element in the similarity matrix $A$ as a learnable scalar as a bias term in the self-attention module, the formula is as follows [23]:

$$A_{ij} = \frac{(h_i W_Q)(h_j W_K)^T}{\sqrt{d}} + b_{\phi(v_i, v_j)}, \tag{4}$$

where $b_{\phi(v_i, v_j)}$ is a learnable scalar indexed by $\phi(v_i, v_j)$ and is shared across all layers.

Compared to traditional GNN, where the receptive field of GNN is confined to the neighborhood, the Transformer layer provides a global receptivity field, and each node can follow all other nodes in the graph. At the same time, nodes in the Transformer layer can adjust their attention to all other nodes through $b_{\phi(v_i, v_j)}$.

### 3.3. Edge Encoding

In graphs, the structural features of edges are also important for graph representation learning, and encoding them into the network is essential. To better encode edge features into the attention layer, the shortest path $SP_{ij} = (e_1, e_2, \ldots, e_N)$ from $v_i$ to $v_j$ for each node pair can be found, and the average of the edge feature embedding representation is calculated to incorporate the edge feature into the attention module as a bias term. The edge encoding formula is [23]

$$c_{ij} = \frac{1}{N} \sum_{n=1}^{N} x_{e_n} \left( w_n^E \right)^T, \tag{5}$$

where $x_{e_n}$ is the feature of the $n$th edge $e_N$ in $SP_{ij}$, $w_n^E \in \mathbb{R}^{d_E}$ is the embedded weight matrix for the nth edge, and $d_E$ is the number of edge feature dimensions.

Therefore, the output after three encodings is [23]

$$A_{ij} = \frac{(h_i W_Q)(h_j W_K)^T}{\sqrt{d}} + b_{\phi(v_i, v_j)} + c_{ij}. \tag{6}$$

After obtaining the similarity matrix $A$, the output $Y$ from the attention module can be calculated, and the calculation formula is as follows:

$$Y = softmax(A)V = softmax\left(\frac{QK^T}{\sqrt{d}} + b_\phi + C\right)V. \tag{7}$$

### 3.4. User Identification Layer

Through the above method, the user representation matrices $Y_S$ and $Y_T$ from the source social network and the target social network can be obtained. The reciprocal of the Euclidean distance $L_2$ is used to measure the similarity matrix $S$ between each user

vector, and the alignment results are obtained according to the similarity. The calculation formula is

$$S_{ij} = \left( \|y_s^i - y_t^j\|_2 + \varepsilon \right)^{-1}, y_s^i \in Y_S, y_t^j \in Y_T, \tag{8}$$

where $\varepsilon$ is a special minimum value.

Different from the traditional cross-social network alignment model, the cross-entropy of any pair of nodes $(v_i, v_j), v_i \in Y_S, v_j \in Y_T$ is not used as the loss function [17] in this paper, because the positive and negative sample size of the association between the two networks is large. Therefore, the method of maximizing the probability of the positive side and minimizing the probability of the negative side as the loss function is adopted. On the one hand, the nodes with link relationships between the two networks are more similar. On the other hand, the distribution of nodes without connection relationships will be more scattered. The calculation formula is

$$L = \text{mean}\left( l_j^i \sum_{(v_i,v_j) \in O_{pos}} \log \eta(S_{ij}) \right) + \text{mean}\left( (1 - l_j^i) \sum_{(v_i,v_j) \in O_{neg}} \log(1 - \eta(S_{ij})) \right), \tag{9}$$

where $\eta$ is the sigmoid function. The value of $v_s^i \in G_S$ and $v_t^j \in G_T$ depends on whether and belongs to the set of anchor users $T$. The formula of $l_j^i$ is

$$l_j^i = \begin{cases} 1, & \left(v_s^i, v_t^j\right) \in T \\ 0, & \left(v_s^i, v_t^j\right) \notin T \end{cases}. \tag{10}$$

## 4. Datasets and Experiments

### 4.1. Datasets

The datasets used in the experiment are two real-world datasets from Cao [40]: Facebook–Twitter and Weibo–Douban. Both datasets are collected from public information in the social network, so there is no privacy breach. Table 2 lists this data and some basic information.

**Table 2.** The information of datasets.

| Dataset | $|V|$ | $|E|$ | $|CV|$ |
|---|---|---|---|
| Facebook | 3481 | 7224 | 1874 |
| Twitter | 3211 | 6020 | |
| Weibo [1] | 1241 | 1625 | 471 |
| Douban [2] | 1170 | 1695 | |

[1] http://www.weibo.com/ (accessed on 31 March 2023), [2] http://www.douban.com/ (accessed on 31 March 2023).

Facebook–Twitter: Facebook and Twitter are both social networking platforms with large numbers of users, and the dataset was collected through third-party platforms dedicated to linking users, collecting a total of 1,107,695 accounts, of which 422,291 accounts were related to Facebook, 669,198 accounts were related to Twitter, and 328,224 pairs of users were associated between the two datasets.

Weibo–Douban: Weibo and Douban are China's largest microblogging sites and movie rating sites, respectively. For the Douban dataset, there are 1,694,399 active users, of which 141,614 are associated with Weibo users. In addition to having social relationships in Weibo's network with Douban, the dataset also collects user-generated content, such as Douban's movie rating history and Weibo's blog history. The average user has 287 blogs and 120 rating histories. Here, we only use the information of social connections.

In Table 2, $|V|$ is the number of users, $|E|$ is the number of edges, and $|CV|$ is the number of associated users in the two social networks.

### 4.2. Comparable Models

DeepWalk: uses random walks to sample the node sequence, and then uses the word2vec model to learn the node embedding.

GCN: extracts features from graph data through GNN for node representation learning, which is widely used in node classification, graph classification, and link prediction.

HGCN: bases on hypergraph convolutional networks for network embedding.

MGCN: combines graph convolutional networks (GCN) and hypergraph convolutional networks to jointly learn network vertex representations at different levels of granularity.

GAT: introduces an attention mechanism based on graph convolution to the weighted summing of the features of neighboring nodes and learning node representation.

### 4.3. Evaluation Metric

To evaluate the performance of the models, we use the most commonly used evaluation metric: accuracy@K (ACC@K), which is defined as

$$ACC@K = \frac{1}{N} \sum_{i=0}^{N-1} A_{v_i}(@k),$$ (11)

where $A_{v_i}(@k)$ indicates whether the source social network user $v_i(v_i \in Y_S)$ corresponds to the user in the target social network $v_i'(v_i' \in Y_T)$ exists in the top $k$ users, and $N$ is the total number of test users in the source social network. In addition, $A_{v_i}(@k)$ is defined as

$$A_{v_i}(@k) = \begin{cases} 1, & v_i' \in topk \\ 0, & v_i' \notin topk \end{cases}.$$ (12)

### 4.4. Analysis of Experimental Results

Tables 3 and 4 show the experimental results on the Facebook–Twitter and Weibo–Douban datasets, showing that the proposed model outperforms the comparable methods on both datasets.

**Table 3.** The performance of different methods on ACC@K on the Facebook–Twitter dataset.

| Dataset | Model | ACC@1 | ACC@10 | ACC@20 | ACC@50 |
|---|---|---|---|---|---|
| Facebook–Twitter | DeepWalk | 0% | 1.32% | 1.97% | 6.58% |
| | GCN | 3.22% | 31.11% | 40.00% | 53.33% |
| | HGCN | 0.89% | 9.43% | 18.87% | 47.17% |
| | MGCN | 1.42% | 10.32% | 20.15% | 58.30% |
| | GAT | 3.77% | 28.89% | 42.22% | 62.22% |
| | TUAM | **8.01%** | **40.50%** | **57.85%** | **78.51%** |

**Table 4.** The performance of different methods on ACC@K on the Weibo–Douban dataset.

| Dataset | Model | ACC@1 | ACC@10 | ACC@20 | ACC@50 |
|---|---|---|---|---|---|
| Weibo–Douban | DeepWalk | 0.21% | 2.34% | 9.55% | 14.23% |
| | GCN | 9.77% | 39.92% | 49.89% | 62.85% |
| | HGCN | 4.25% | 30.15% | 47.56% | 59.45% |
| | MGCN | 5.31% | 31.85% | 50.32% | 63.69% |
| | GAT | 11.25% | 37.37% | 52.65% | 66.88% |
| | TUAM | **12.06%** | **53.90%** | **65.25%** | **90.07%** |

In this table, the ACC@K, K = 1, 10, 20, and 50 of the existing and proposed user alignment models on the Facebook–Twitter dataset are compared. The results show that the proposed TUAM performs better than other models.

According to Tables 3 and 4, it is illustrated that the TUAM model outperforms other comparable models in the two datasets. When K equals to 10 or 20, the performance of

TUAM has the most significant improvement compared to other models. When K = 10, the accuracy rate ACC@10 in Facebook–Twitter and Weibo–Douban improved by 11.61% and 16.53% compared to GAT, respectively. The DeepWalk model embeds nodes by random walk, but this process does not map the node features of the two networks to the same vector space, but directly performs user alignment, which severely reduces the accuracy of the model. GCN model can effectively aggregate the surrounding neighbor information, but it does not consider the importance of each neighbor node, which limits the learning ability of the features of network nodes. HGCN and MGCN are based on GCN, which greatly benefits from the nonlinearity of neural networks, but the modeling of hyperedge is very redundant for non-hypergraph problems, which not only has no benefit to the model, but also reduces the accuracy of the model. The attention mechanism in GAT perfectly solves the problem of GCN, making GAT perform better than GCN in the two datasets, but it is still limited by the "over-smooth" problem and cannot mine the characteristics of network nodes at a deeper level. However, the three encodings of TUAM can learn network structure information as well as GNN, or even better, and will not be limited by the "over-smooth" problem, can mine network node features at a deeper level, and TUAM's global receptive field makes the model learn higher-level structural features better, so it performs better than other models. The accuracy rates of these models on two datasets are shown in Figures 2 and 3.

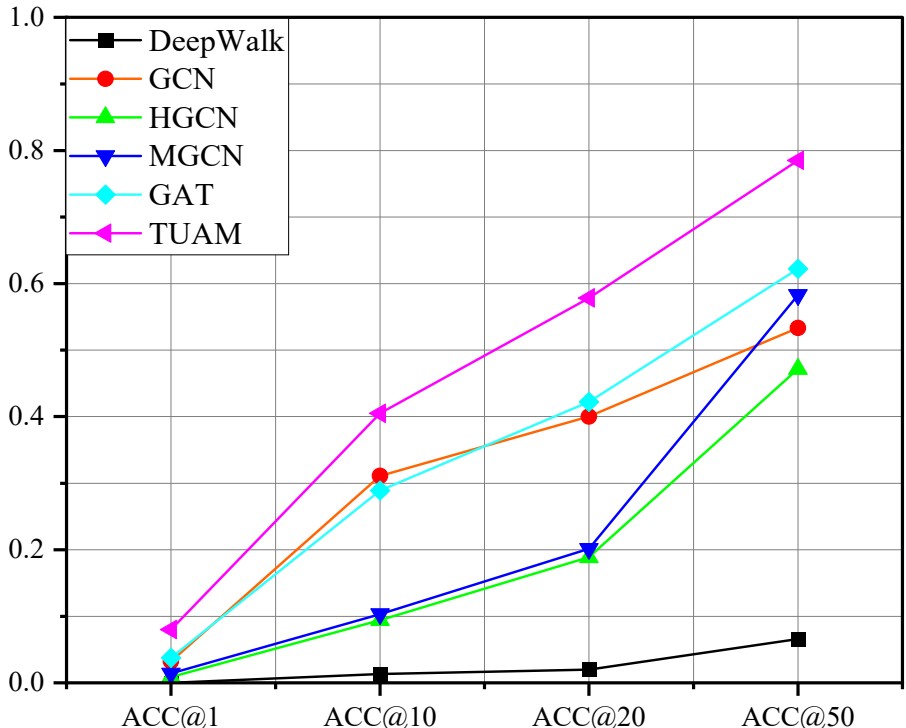

**Figure 2.** The accuracy of several models on the Facebook–Twitter dataset.

In Figures 2 and 3, it is obvious that the accuracy of TUAM on two datasets is higher than other existing models, especially when K = 50. The ablation experiments on the importance of the three encodings are conducted in TUAM across the Facebook–Twitter and the Weibo–Douban datasets. The ablation results are shown in Tables 4 and 5. The best results are indicated in bold font.

According to Tables 5 and 6, the experimental results show that for the TUAM without centrality coding, spatial coding, or edge coding, the effectiveness of the TUAM decreases. This is because the centrality encoding module can effectively encode the information of different nodes into Transformers to improve the accuracy of model recognition. At the same time, the spatial encoding module and edge encoding can effectively capture

the spatial information and structural information of nodes, which is more conducive to the expression of the structural characteristics of the Transformer learning network. The three kinds of encoding methods can effectively convert the graph topology information into sequence data information that is conducive to Transformer learning. TUAM does not need to consider the "over-smooth" phenomenon caused by too-deep layers, such as GCN, and has a larger global receptive field, efficiently learns topology structure features, and improves model performance. The trends of accuracy on Facebook–Twitter and Weibo–Douban datasets of ablation experiments are shown in Figures 4 and 5.

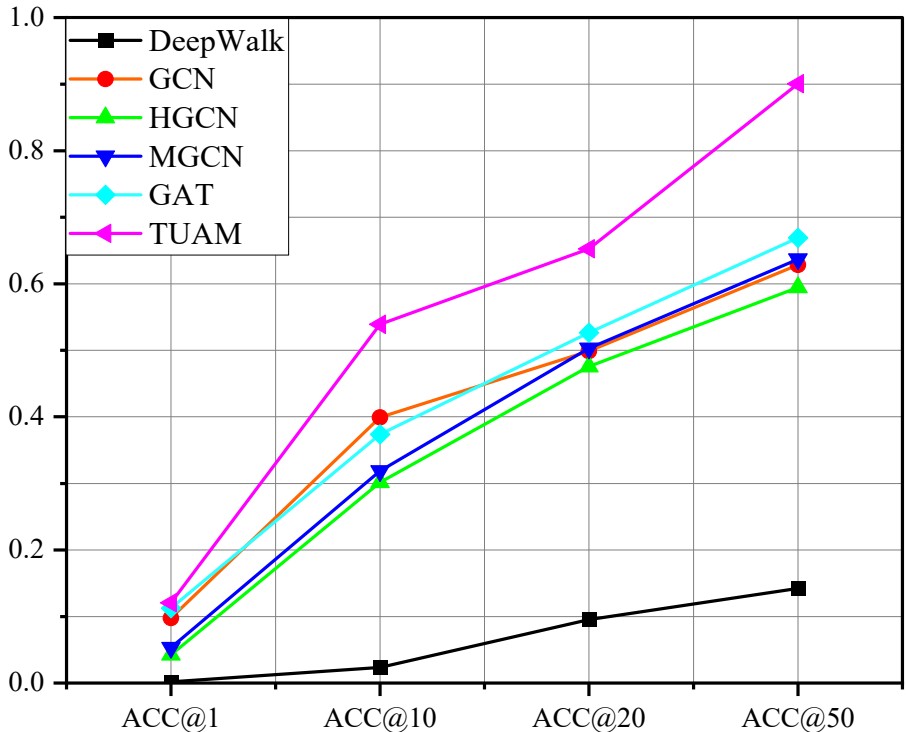

**Figure 3.** The accuracy of several models on the Weibo–Douban dataset.

**Table 5.** Ablation experiment results on the Facebook–Twitter dataset.

| Centrality Encoding | Spatial Encoding | Edge Encoding | ACC@1 | ACC@10 | ACC@20 | ACC@50 |
|---|---|---|---|---|---|---|
| × | √ | √ | 6.94% | 35.41% | 48.40% | 72.16% |
| √ | × | √ | 7.65% | 32.56% | 46.62% | 68.86% |
| √ | √ | × | 6.58% | 32.92% | 47.86% | 68.51% |
| √ | √ | √ | **8.01%** | **40.50%** | **57.85%** | **78.51%** |

**Table 6.** Ablation experiment results on the Weibo–Douban dataset.

| Centrality Encoding | Spatial Encoding | Edge Encoding | ACC@1 | ACC@10 | ACC@20 | ACC@50 |
|---|---|---|---|---|---|---|
| × | √ | √ | 7.09% | 51.77% | **67.38%** | 85.80% |
| √ | × | √ | 9.22% | 48.94% | 63.12% | 83.69% |
| √ | √ | × | 6.38% | 41.84% | 63.83% | 78.72% |
| √ | √ | √ | **12.06%** | **53.90%** | 65.25% | **90.07%** |

It is shown that TUMA with three encoding methods has better performance on accuracy on two datasets compared to the model without one of the three encodings.

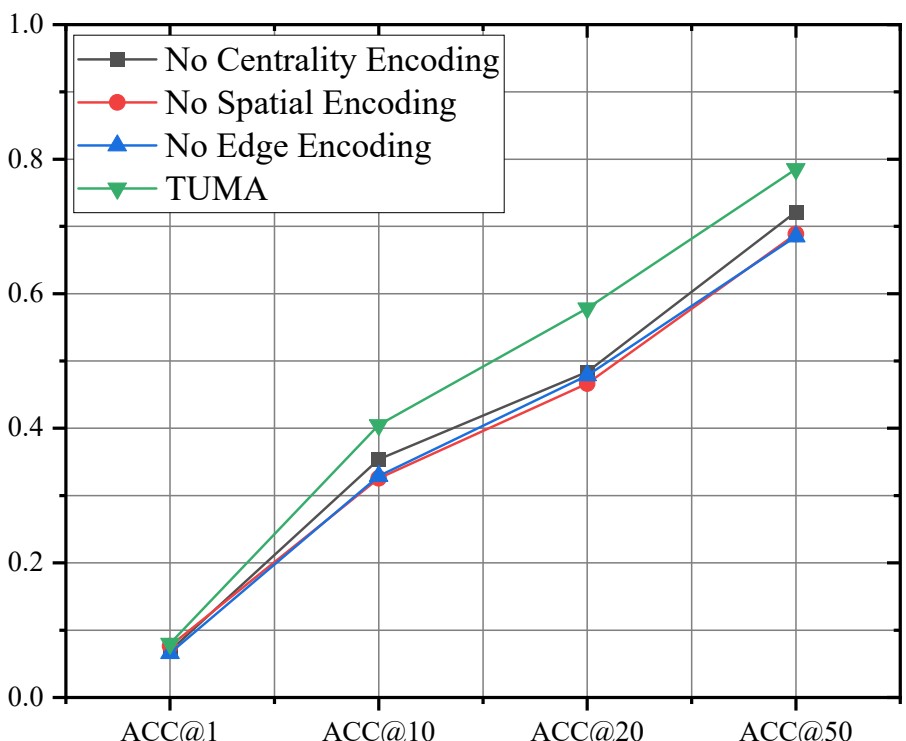

**Figure 4.** The accuracy of ablation experiments on the Facebook–Twitter dataset.

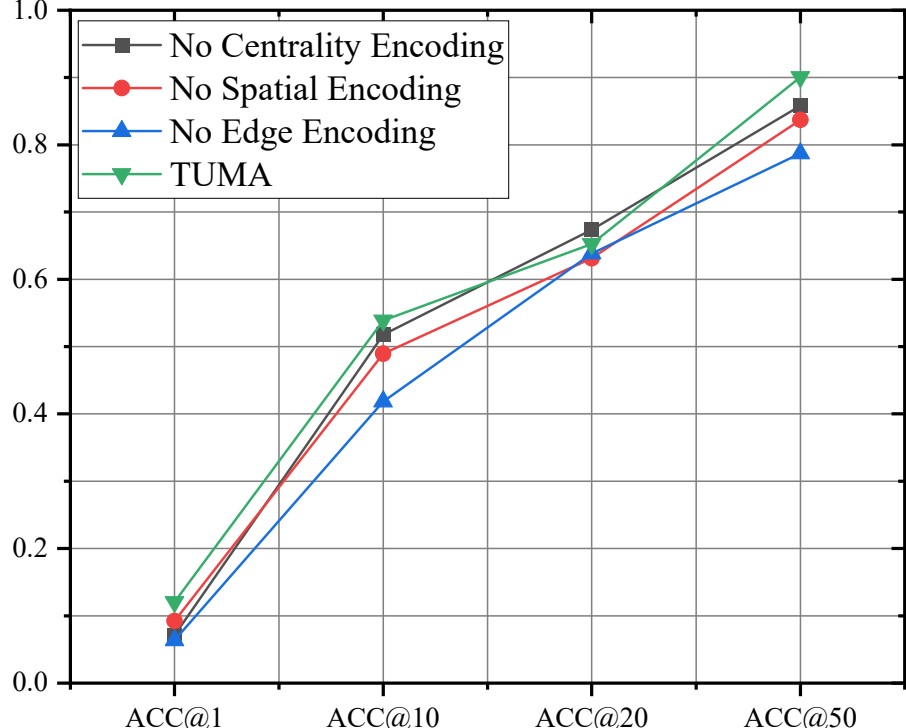

**Figure 5.** The accuracy of ablation experiments on the Weibo–Douban dataset.

## 5. Conclusions

In this paper, user alignments across social networks were described. The research received a lot of attention in both academia and industry, and was involved in many social network-related applications, such as link prediction, interest recommendation, etc. A Transformer-based user alignment model based on network topology information was proposed to learn the structural information between nodes in the networks. The proposed model is different from the traditional GCN through graph convolution to obtain network structure information from neighbor nodes, but through a specific encoding method to express the graph structure information in the form of sequence data. Experiment results show that the proposed method can better describe the association relationship between node neighbors, has a more accurate vector representation of nodes, and can improve the accuracy of user association matching.

While our approach has certain advantages, there are also some drawbacks. Our work only makes use of the structural information of the network, which is less informative. If additional attribute information is considered, it will be helpful to improve the performance of the model. At the same time, the proposed model cannot be adapted to large-scale graph datasets. Therefore, the future research direction is to build a framework for integrating social network structure information and attribute information on the basis of this user alignment model. In addition, it is necessary to reduce the cost of model calculation to make it suitable for large-scale alignment across social networks.

**Author Contributions:** Conceptualization, L.J. and S.L.; methodology, G.W. and F.P.; software, L.W. and T.L.; validation, T.L. and S.L.; investigation, T.L., L.W., and G.W.; writing—original draft preparation, T.L.; writing—review and editing, G.W. and F.P.; funding acquisition, L.J. and S.L. All authors have read and agreed to the published version of the manuscript.

**Funding:** This research was funded by Songshan Laboratory Project, Major Science and Technology Project of Henan province, No: 221100210700-2.

**Data Availability Statement:** Not applicable.

**Conflicts of Interest:** The authors declare no conflict of interest.

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
