# Peer review of "Transformer-Based User Alignment Model across Social Networks"

_electronics, doi:10.3390/electronics12071686_

Round 1

Reviewer 1 Report

Dear Authors

Paper title: Transformer-based User Alignment Model Across Social Net 2 works

Manuscript ID: electronics-2314198

Although the study is promising, there are some concerns regarding the details of the presented study as well as the organization of paper. Furthermore, the structure of the submitted manuscript and the literature survey requires some improvements.

Here are some comments and suggestions to improve the paper. This comments are based on a number of points that I encountered while reading the paper which I will list the most important ones:

1.      Please avoid using the word “I”, “We” in the technical paper.

2.      Kindly include some statistical data about the proposed the model and its growth in the abstract.

3.      In did not find any comparison of the proposed model with the existing model. Kindly provide the proper comparison, to prove the effectiveness of the system

4.      I have not seen experimental results in the paper, but is it mentioned in the abstract. kindly check

5.      I feel the word “transformer” can newly framed, as the transformer is the universal thing which is used for step up/ step down the voltage.

6.      In the manuscript, there are several equations, but is not clear, why they are mentioned.

7.      What is Q, K, V in expression equation 1? kindly abbreviate it

8.      I don’t find any latest reference in the manuscript. kindly include the latest reference of the total paper, 30% should be from last 3 years and 50% from last 5 years.

9.      Cite the equations, if it is directly copied from the other paper.

With regards

Reviewer. 

Reviewer 2 Report

To learn the structural information between nodes in networks, the authors of this paper propose a transformer-based user alignment model based on network topology information. The authors contend that their suggested model differs from the conventional GCN in that it expresses graph structure information as sequence data using a particular encoding technique. The proposed method, which has a more accurate vector representation of nodes and enhances the accuracy of user association matching, can better describe the association relationship between node neighbours, according to the authors' analysis of the experiment results.

My observations are listed below:

1.     Although you assert that your model is superior, I would like to know how you came to this conclusion. If you already knew it, did you forget to include that information in the paper's description? The literature review informs the authors on this side as well as the readers. It details what has already been done and what advancements the new research calls for. As a result, I would ask for a thorough literature review and comparison table to show readers how your work differs from and advances the state-of-the-art.

2.     How did you discover that the algorithms and methods you used and applied were the ones you needed to use and apply? The literature review, which informs you of the gaps in the current state-of-the-art, also serves as a guide in this. Therefore, you must justify the precise methods that you have used in this paper.

3.     The list of study limitations is not entirely complete.

4.     You need to be a little more specific about the outside source who assisted you in gathering Facebook, Twitter, and Weibo data.

5.     Which attributes/columns of data did you choose and which ones did you reject? Please specify.

Reviewer 3 Report

This paper is well-written and addresses a significant problem: user alignment across social networks. The authors proposing to use Transformer to solve user alignment across networks presented an interesting use case of a Transformer-based neural network, which is currently a hot topic in the field of deep learning. The methodology is clearly explained, and results are extensively discussed. 

Author Response

Thanks for your comments.

Round 2

Reviewer 1 Report

Dear Author

Paper is well revised, it may be accepted in current form. 

Reviewer 2 Report

I'm satisfied with your modifications in the paper.

Thanks